# Dietary Supplementation of Eubiotic Fiber Based on Lignocellulose on Performance and Welfare of Gestating and Lactating Sows

**DOI:** 10.3390/ani13040695

**Published:** 2023-02-16

**Authors:** Agnês Markiy Odakura, Fabiana Ribeiro Caldara, Maria Fernanda de Castro Burbarelli, Ibiara Correia de Lima Almeida Paz, Rodrigo Garófallo Garcia, Viviane Maria Oliveira dos Santos, Daniella Ferreira de Brito Mandu, Jaqueline Murback Braz, Marconi Italo Lourenço da Silva

**Affiliations:** 1College of Agrarian Sciences (FCA), Federal University of Grande Dourados (UFGD), Dourados 79804-970, Brazil; 2Faculty of Veterinary Medicine and Animal Sciences (FMVZ), São Paulo State University (UNESP), Botucatu 18618-970, Brazil; 3Faculty of Veterinary Medicine and Animal Sciences (FAMEZ), Federal University of Mato Grosso do Sul (UFMS), Campo Grande 79070-900, Brazil

**Keywords:** behavior, infrared thermography, intestinal microbiota, reproductive performance, well-being

## Abstract

**Simple Summary:**

The addition of fiber to the diet of sows can improve welfare due to prolonging the satiety feeling, which contributes to reducing stereotyped behaviors and providing benefits to reproductive performance. Thus, the present study was carried out aiming to evaluate the effects of lignocellulose-based eubiotic fiber supplementation on the behavior, surface temperature, and reproductive parameters of sows during pregnancy and lactation, as well as the performance of their litters. Dietary eubiotic fiber supplementation for sows in the final third of pregnancy and lactation improves their welfare and performance of their piglets.

**Abstract:**

The present study aimed to evaluate the effects of partially fermentable insoluble dietary fiber supplementation on the behavior, surface temperature, and reproductive parameters of gestating and lactating sows, as well as on the performance of their litters. Four hundred hyper-prolific sows were assigned in a randomized block design with two treatment groups during the gestation phase: Control (C), corn-soy based diets, or corn-soy based diets with daily 55 g supplementation of eubiotic fiber (F) from the 85th day of gestation until the farrowing (late pregnancy). During the lactation phase, the sows were assigned in a 2 × 2 factorial design using the following treatment groups: (1) CC, no fiber included during gestation and lactation. (2) FC, daily inclusion of 55 g of fiber only during late pregnancy. (3) CF, daily inclusion of 55 g of fiber only during lactation. (4) FF, daily inclusion of 55 g of fiber during late pregnancy and lactation. Sows that received dietary fiber supplementation during the final third of gestation increased feed intake during lactation. Piglets from sows supplemented in both phases showed a significant increase in weight at weaning. The dietary supplementation of eubiotic fiber for sows in the end period of gestation and lactation improved performance and welfare, with positive consequences for developing their litters.

## 1. Introduction

The evolution of the genetic improvement of pigs in recent decades has promoted a low average weight per piglet at birth, less access to colostrum, and low availability of productive teats [1]. In addition, current sows have a higher body weight and a lower pattern of voluntary feed consumption. These factors may be related to the reduction in the survival rates of piglets during lactation, which results in low weaning weight and low litter uniformity [2].

Due to the great number of piglets, farrowing has become longer with a greater probability of depleting the sow’s energy reserves, impairing farrowing kinetics, and predisposing piglets to hypoxia [3]. Therefore, the supply of dietary and partially fermentable fibers in the diet, aiming to increase the energy supply to females during parturition, a time of high energy demand, might be an advantageous nutritional strategy at the end of gestation [4,5]. The use of partially fermentable fibers during food restriction phases, such as gestation, shows benefits related to the promotion of immediate and longer satiety [6] and modulates the microbiota, which favors intestinal health and reduces digestive disorders [7]. Studies have shown that combining fermentable and non-fermentable fractions of fiber in diets provides physiological benefits to the animals as its prebiotic functions [8,9] can maintain a balanced microflora, ensuring intestinal eubiosis and, thus, improving animal performance [10,11,12]. These fiber effects can play a key role in feeding strategies that seek to reduce the use of antimicrobial growth promoters.

In addition to gastrointestinal and welfare benefits, high-fiber diets during gestating in sows are related to increased voluntary feed intake during lactation due to the residual effect on feed intake capacity [13] and improved performance of suckling piglets [14]. Increasing sow food consumption during lactation is a challenge, especially in tropical climate conditions, where there is a significant reduction in feed intake, a decrease in milk production, high body catabolism, and a reduction in litter size and weight [15]. Therefore, this practice needs to be better evaluated in hot climate regions, considering that during the fermentation process and absorption of their products, fibers generate a caloric increment and increase the visceral mass of animals, which also increases metabolic heat [16].

Therefore, the present study aimed to evaluate the effects of partially fermentable insoluble dietary fiber supplementation on the behavior, surface temperature, and reproductive parameters of gestating and lactating sows and the performance of their litters.

## 2. Materials and Methods

### 2.1. Animal Ethics Statement

All procedures performed in this study were approved by the Ethics Committee on the Use of Animals (CEUA) at the Federal University of Grande Dourados, under protocol no. 04/2021.

### 2.2. Site Description

The experimental protocol was conducted in a commercial piglet production unit (Weaned Piglet Production) hosting 2689 sows. The site is located in Ivinhema, Mato Grosso do Sul, Brazil (22°21′45″ S, 53°52′49″ W) at 406 m in altitude. The climate in the region, according to the Köppen classification, is Aw, i.e., tropical climate with dry winters, with an annual rainfall average between 1200 and 1800 mm and an annual temperature average of 25 °C, possibly reaching highs up to 40 °C in spring and lows of 10 °C in winter.

### 2.3. Animals and Facilities

Four hundred DanBred sows from the 85th day of gestation to the weaning of piglets (at 20 days of age) were used. During the gestation phase, the sows were housed in 20 collective pens in a masonry barn equipped with fans and water sprinklers. All pens contained solid floor (2/3) and slatted floor (1/3), nipple drinkers, and automatic feeders (drops). Twenty sows were housed per pen, with a stocking density of 2.0 m^2^ per sow. Seven days before the expected day of parturition, the sows were transferred to the farrowing barn and housed in conventional farrowing crates (2.30 mL × 0.80 mW) equipped with a feeder, a nipple drinker, a creep heated with an incandescent lamp, and an escape area for piglets (0.45 mW). After the weaning, the sows returned to the gestation barn and were housed in individual crates, where they were monitored to determine the weaning-to-estrus interval (WEI). All animals (sows and piglets) had *ad libitum* access to water. House temperature (°C) and relative humidity (%) were recorded daily using thermo-hygrometers (Novo Test TH802A thermo-hygrometers, São Paulo, Brazil) positioned in the middle of the pens and cages.

### 2.4. Experimental Design and Treatments

At 85 days of gestation (late pregnancy), 400 sows were distributed in four blocks according to parturition order (PO1, PO2, PO3, and PO > 3) and randomly allocated in two treatments: Control (C), corn-soy-based diets, or corn-soy-based diets with daily 55 g supplementation of eubiotic fiber (F). At 107 days of gestation, the sows were transferred to the farrowing room and assigned in a 2 × 2 factorial design with the following treatments:(1)Control-Control (CC): Corn-soy-based diets without supplementation of eubiotic fiber during late pregnancy and lactation.(2)Fiber-Control (FC): Corn-soy-based diets with daily 55 g supplementation of eubiotic fiber during late pregnancy and corn-soy-based diets during lactation.(3)Control-Fiber (CF): Corn-soy based diets during late pregnancy and corn-soy-based diets with daily 55 g supplementation of eubiotic fiber during lactation.(4)Fiber-Fiber (FF): Corn-soy-based diets with daily 55 g supplementation of eubiotic fiber during late pregnancy and lactation.

During the gestation period, the sows were housed in collective pens, and the pen (*n* = 10) with 20 sows each was considered as an experimental unit. After the transfer to the farrowing room and subdivision into the 4 treatments, each sow came to be considered the experimental unit (100 replicates per treatment).

### 2.5. Experimental Feed and Fiber Supply

The corn-soy-based experimental diets provided in the final third of gestation (late pregnancy) and during the lactation phase were formulated to meet the nutritional requirements of each phase (Table 1). The eubiotic fiber used comprised lignocellulose (30% lignin) derived from selected and treated fresh wood, 100% insoluble and partially fermentable with a higher crude fiber content (65%) compared to traditional fiber sources and free mycotoxins. A low inclusion rate (0.5–3%) was used.

During the gestation phase (from 85 to 106 days), the diets were provided twice a day (2.2 kg/sow/day) at 06:00 and 08:00 h using automated drops. The sows from the F treatment were supplemented with eubiotic fiber (55 g/sow/day) diluted in 445 g of cookie meal (used as a vehicle). After dilution, the amount was divided into two equal parts and supplemented on-top of the feed. Following the same feeding management, the sows from the C treatment received the same cookie meal amount without the eubiotic fiber.

After being transferred to the farrowing facilities (at 107 days of gestation) and before farrowing, the sows began receiving 3.2 kg of lactation diet divided into two meals a day at 09:00 and 15:00 h. According to the treatments, sows were supplemented with 55 g of eubiotic fiber plus vehicle or only vehicle at the feeding time. On the day scheduled for farrowing (at 114 days of gestation), the sows did not receive feed, according to the management adopted by the farm. When the sows showed prepartum signs, such as edema and secretion of the vulva (approximately 114 days of gestation), the parturition was induced by intramuscular injection of prostaglandin associated with oxytocin (5 mg) in the first application and after six hours for the second application (5 mg), according to the farm’s protocol.

After farrowing, the feed supply was resumed and divided into four meals (8.0 kg/sow/day) at 06:00, 09:00, 15:00, and 21:00 h. According to the treatments, sows were supplemented with 55 g of eubiotic fiber plus vehicle or only vehicle in the second feeding time.

### 2.6. Measurements

#### 2.6.1. Body Weight, Feed Intake, and Body Condition Score (BCS)

Sows were individually weighed on a mechanical scale (Lider Balances B650G/LD1050, São Paulo, Brazil), with a capacity of 500 kg when they were transferred to the farrowing room (at 107 days of gestation) and at weaning (20 days after farrowing). Feed consumption and leftovers in feeders were also recorded daily during the experiment to calculate average daily feed intake (ADFI). Due to feed restriction in the gestation phase, no feed leftovers were recorded.

The body condition score was evaluated concomitantly with the body weight. The Caliper equipment (Mitutoyo Digimatic Caliper, São Paulo, Brazil) was used to objectively quantify the angularity on the sows’ back at point P2 (6.5 cm from the dorsal midline posterior to the last rib). Thus, the sows’ body score was measured indirectly and classified as 1 (lean), 2 (ideal), or 3 (fat) [17].

#### 2.6.2. Reproductive Performance

The number of piglets born alive and stillbirths, duration of farrowing (from the birth of the first piglet to delivery of the placenta), number of weaned piglets, and weaning-to-estrus interval were recorded to evaluate reproductive performance. After weaning, the sows were housed in individual pens in the gestation barn, marked, and monitored daily until entering estrus. The estrus was considered when the sow showed the male tolerance reflex. Estrus diagnosis was performed twice a day at 07:00 and 17:00 h.

#### 2.6.3. Litter Performance

All delivered piglets (alive and stillborn) were individually weighed during farrowing using a digital electronic scale (Prix Toledo BS20, São Paulo, Brazil). Following the standard handling procedure of the farm, the litter was standardized to balance the number of piglets per sow and to equalize the weight of piglets. Fourth-eight hours after standardization of litters, performed only between sows of the same treatment, all piglets were weighed again. At weaning (20 days of age), all piglets were individually weighed. The coefficients of variation of the piglets’ weights (at birth and weaning) were calculated to assess litter uniformity plus the number of piglets born weighing less than 1000 g.

### 2.7. Sows Surface Temperatures

An infrared thermographic camera (Flir Studio, CATS60 Pro, São Paulo, Brazil) was used to measure sows’ body surface temperatures (MST) during gestation and lactation and the mammary gland surface temperatures during lactation. The images were taken once a week at 08:00 h and 15:00 h. Thermographic images were read by converting the color spectrum into surface temperature using the Flir software (IRSoft, version 3.6, Testo Thermal Imagers, Wilsonville, OR, USA). The emissivity coefficient used was 0.96 for the entire sow body surface. The mean surface temperature and the standard deviation of body area were calculated using the temperature of 30 evenly distributed points to represent the sows’ and mammary apparatus’s global body surface (Figure 1).

### 2.8. Behavior

Behavior assessments were performed using images captured by 8 video cameras installed at a 2 m height that allowed a wide view of pens (gestation phase) and farrowing crates (lactation phase). The images were recorded and stored on an external hard drive to be analyzed later.

The behavior of sows in the gestation phase, from the 85th day of gestation until their transfer to the farrowing barns (at 107 days of gestation), was evaluated once a week by the scanning method, with observation at 10 min intervals from 06:00 to 17:00 h, at pen level. A number of 20 sows each were randomly selected and recorded. The sum of behaviors totaled 100% of the compiled assessments.

The behavior of sows in the lactation phase was evaluated once a week by the focal animal method at ten-minute intervals, from 06:00 to 17:00 h, at sow level, totaling 67 observations per sow.

Pre-established ethograms were used (Table 2 and Table 3) according to the reproductive cycle stage.

### 2.9. Statistical Analysis

The statistical assumptions of normality of residuals and homogeneity of variances of the reproductive performance data and body temperature of sows were verified by Shapiro Wilk and Levene tests, respectively. Variances that met the assumptions were subjected to analysis of variance using the SAS MIXED procedure (Version 9.4, SAS Institute Inc., Cary, NC, USA). The variables that did not meet the assumptions (number of stillbirths, mummified piglets, calving time, body condition score, estrus weaning interval) were transformed using the LOGNORMAL matrix and the GLIMMIX procedure, which models the logarithm of the response variable as a normal random variable. In both procedures in the mathematical model, parturition order and ambient temperature were added as covariates for the variables of productive performance and body temperature, respectively. The effects of interaction between the factors were verified when significant and unfolded. When using the MIXED procedure, the effects of unfolding were evaluated by the F test. In the GLIMMIX procedure, to compare the means by the least squares test, the obtained estimates were adjusted by the inverse link (pdiff ilink lines) of the GLIMMIX procedure.

The statistical analyses for behavioral results were performed using the SAS GLIMMIX procedure (SAS, version 9.4, SAS Institute Inc., Cary, NC, USA). As they did not meet the assumption of normality, the residue was transformed using the LOGNORMAL matrix. Thus, the GLIMMIX procedure modeled the logarithm of the response variable as a normal random variable. The mean and the variance were estimated on the logarithmic scale, thus assuming a normal distribution. Behavioral assessments were performed on more than one occasion on different days. The effect of time of assessment was added to the mathematical model as a covariate. Thus, an analysis of variance was performed using the PROC GLIMMIX, evaluating the effect of fiber inclusion in the diet during gestating and the effects of interactions between dietary fiber supply during gestating and during lactation. The statistical model for the gestation phase included the treatments as a fixed effect, the sows nested in the groups as random effects, and the sampling day as a covariate. The statistical model for parturition/lactation included treatments as a fixed effect and day of sampling as a covariate. To compare the means by the least squares test, the obtained estimates were adjusted by the inverse link (pdiff ilink lines) of the GLIMMIX procedure, followed by the F-test, and assigned significance when (*p* < 0.05).

### 2.10. Cost–Benefit Analysis of Using Eubiotic Fiber

In order to evaluate the cost–benefit of using eubiotic dietary fiber, the costs per kg of weaned piglets produced were calculated considering feeding costs (feed and dietary fiber consumption) during gestation and lactation, the weight of piglets at weaning, and the average number of piglets weaned per litter. The cost of eubiotic fiber was determined in a commercial consultation at a value of USD 1.35/kg. The costs of the gestation and lactation foods were calculated at USD 0.43/kg and USD 0.54/kg, respectively. The price of weaned piglets was USD 2.83 per kg live. At the time of calculations (12 December 2022), the quotation of BRL 5.29 (Real—Brazil’s national currency) per dollar was used.

## 3. Results

### 3.1. Reproductive Performance and Litter Performance

There was no effect of eubiotic dietary fiber supplementation for sows on weight at farrowing and weaning-to-estrus interval (*p* > 0.05). The mean weight loss of sows between prepartum and weaning was 26.96 kg, and the mean weaning-to-estrus interval was 4.7 days (Table 4).

Considering the litter standardization management carried out in the first 48 h after birth, the distribution was balanced. Therefore, there was no difference in the number of piglets per sow and their weight (W 48 h) between different treatments (*p* > 0.05) that could affect the results obtained (Table 5). There was no effect of treatments on the control number of piglets weaned, coefficient of variation of piglet weight at weaning, and mortality rate of suckling piglets (*p* > 0.05).

The supply of eubiotic dietary fiber to sows during the final third of gestation (late pregnancy) did not affect the number of piglets born alive, stillbirths, average piglet birth weight, litter uniformity, number of piglets born with less than 1.0 kg, weight, and BCS of sows at the entrance to the farrowing unit (*p* > 0.05) (Table 6). Farrowing duration was shorter for sows from the FF treatment (*p =* 0.006).

Piglets from sows that received fiber supplementation during lactation (CF and FF) were weaned heavier than those whose sows did not receive fiber (CC) or received fiber only during late pregnancy (FC). However, piglets whose sows received fiber in both periods were also heavier than piglets from sows that only received fiber during lactation, showing an additive effect of fiber during both phases on litter performance (*p* = 0.003) (Table 4). Sows supplemented in late pregnancy with eubiotic dietary fiber showed a higher feed intake during the lactation phase regardless of the continuation of supplementation at this phase (*p* < 0.05). In addition, fiber intake during the lactation phase promoted a better body condition score at weaning (Table 4).

### 3.2. Surface Temperature of Sows

Sows from treatments that received fiber supplementation during late pregnancy had a higher body surface temperature (34.94 °C) compared to those that did not, regardless of whether they were receiving or not fiber during lactation (Table 7).

On the other hand, sows that received fiber in both phases had a higher temperature of breast apparatus (34.88 °C) than those that received fiber only during gestation (34.35 °C). Fiber supplementation did not affect the body surface temperature of sows during the gestation phase in both periods evaluated (morning and afternoon) (Table 8).

### 3.3. Behavior

There was no effect of fiber supplementation on lateral lying, sitting, kneeling, standing, walking, eating, drinking, nosing, and negative interaction (Table 9). Sows from the F treatment spent more time lying ventrally and less time interacting positively with their peers than sows from the C treatment. In addition to reducing the frequency of stereotyped behaviors such as floor licking and false chewing, sows from the F treatment showed a trend to reduce negative interactions than sows from de C treatment (*p* = 0.064).

Sows that received fiber in only one of the phases of the production cycle (CF and FC) spent less time lying on their sides compared to those that received fiber in both phases (FF). Similarly, sows that did not receive fiber supplementation during gestation or lactation showed a lower frequency of ventrally lying behavior compared to those that received fiber in both periods (Table 10). Sows that received fiber during lactation, regardless of whether they received it during gestation, spent less time sitting and showed a more frequent breastfeeding behavior, in relation to the sows that did not receive fiber during this period.

Sows that received fiber during gestating showed a higher frequency of eating during lactation, even if they were not being supplemented at this stage, than sows from the CC and CF treatments. Additionally, sows that received fiber during lactation (regardless of having been supplemented in the previous period) spent more time eating than those that did not. Sows that received dietary fiber supplementation during gestation and lactation showed more drinking behavior than those who received fiber only during lactation.

The dietary supplementation during both phases (FF) reduced false chewing and biting rails of the farrowing units. However, sows that received fiber during gestation, but did not receive fiber during lactation, showed high levels of stereotypies, which could demonstrate the momentary effect of fiber in improving the feeling of satiety (Table 8).

### 3.4. Cost–Benefit Analysis of Using Eubiotic Dietary Fiber

The cost–benefit evaluation of using eubiotic dietary fiber showed an advantage for the group in which the supplementation was performed in both phases (FF) compared to the other groups. The use of fiber only during the gestation phase was not feasible, as it increased the cost per kg of weaned piglets and consequently reduced the profit obtained per litter (Table 11).

## 4. Discussion

### 4.1. Performance and Reproductive Indexes

The supplementation of eubiotic fiber for sows in the final third of gestation promoted an increase in feed intake of approximately 25.80 kg per sow during the lactation phase. Sows were fed restrictively during gestation to avoid excessive body weight gain. On the other hand, they must consume feed ad libitum during lactation to meet nutritional demands, maximize milk production, and minimize body catabolism [19]. However, promoting a sudden increase in feed consumption by lactating females (from 1.8 to 2.2 kg to 6.0 to 8.0 kg of feed/sow/day) is a great challenge, especially in tropical climatic conditions, where there is a significant reduction in feed intake [20].

Feeding high-fiber diets for gestating sows has been linked to increasing voluntary feed intake during lactation and improving suckling piglets’ performance [19,20]. However, the mechanisms related to the effects of consumption of diets with fiber inclusion during gestation on the consummatory behavior of sows during lactation have not yet been fully explained. Possible explanations have been considered, such as gastric dilatation and an increased total capacity of the gastrointestinal tract [[18],[19],[21],[22],]. Researching the effects of using fiber in diets for pregnant sows, ref. [23] observed that feed formulated with 5% fermented soy fiber had a swelling capacity of 1.89 mL/g, which means that the total volume of diet for gestating sows may practically be double, thus stimulating the increase in capacity of voluntary ingestion and, consequently, satiety.

The amount and nature of fibrous carbohydrates incorporated in swine diets can influence the time it takes for food to travel throughout the gastrointestinal tract. Insoluble fibers remain intact along the gastrointestinal tract, requiring mechanical action to stimulate peristalsis. Thus, it promotes greater motility of the digesta, accelerates the passage rate, and increases food consumption [24].

According to [19], sows fed on a high-fiber diet soluble and insoluble during gestation had lower leptin concentrations before farrowing, which were negatively correlated with feed intake during lactation.

Sows supplemented with eubiotic fiber during gestation showed better farrowing kinetics, resulting in shorter farrowing compared to sows that were not supplemented during pregnancy. Feed restriction on the farrowing day is commonly adopted in many production systems, aiming to prevent constipation, narrow the birth canal if the gastrointestinal tract is full, and reduce the elimination of feces to avoid farrowing crate contamination [25]. Insoluble fibers promote improvements in intestinal motility, preventing constipation during farrowing [26]. In addition, the feed consumption restriction in the last hours before farrowing may lead to a reduction in the concentration of serum glucose and consequently to less energy available for the uterine and muscle contractions necessary for the expulsion of the fetus [27]. Therefore, the time interval between the last meal and the beginning of parturition is crucial for the female to have the energy needed for this event [28]. Studies have shown that the gastrointestinal tract absorbs glucose between four and six hours after feed consumption [29,30], which is directed to tissues and organs, and its serum levels tend to decrease rapidly after insulin secretion [31]. According to [25], sows that consumed feed within three hours before farrowing had higher blood glucose levels, resulting in shorter farrowing times (3.8 h) and lower mortality rates of piglets compared to sows that received food six hours before the start of labor (9.3 h of labor duration).

The provision of fiber-rich diets to sows promotes lower postprandial glycemic concentration, making it stable for a long period and contributing to the sow’s satiety and adequate energy supply during farrowing. Thus, it is possible to restrict the food supply in the last hours before farrowing, ensuring an energy level from glucose oxidation capable of supporting uterine contractions [25,32,33].

When comparing the supply of diets formulated with low fiber (LF), high soluble fiber (HF-S), or high insoluble fiber (HF-I) to gestating sows, [29] observed that consumption of the LF diet resulted in a rapid increase and absorption of glucose from zero to four hours after feeding, while the HF-I and HF-S diets promoted a pattern of glucose absorption at a reduced rate.

Intestinal fiber fermentation promotes the synthesis of short-chain fatty acids (SCFA), which contributes to selecting a more efficient microbiota. This positive modulation of the microbiota, with high production of SCFA, helps to stabilize interprandial glucose, providing late and prolonged glycemic peaks [34]. Thus, converting into an energy source when the glucose supply is in the intestine is insufficient for the sow to develop her activities [35].

Short farrowing reduces the risk of premature umbilical cord rupture and oxygen deprivation to the fetus. Prolonged asphyxia in the uterus may result in the death of the fetus during farrowing [35]. Still, it can make piglets take a long time to find the udder and ingest the colostrum, making them less capable of surviving in extrauterine life [36,37]. In addition, the duration of farrowing can be a risk factor for the health status of sows [38].

In this current study, the eubiotic fiber supplementation did not positively affect the number of piglets born alive and the birth weight. This could be associated with the supplementation only in the final third of gestation, thus not affecting embryonic survival. Previous studies have reported that the positive effects of high-fiber diets on litter size and birth weight are more visualized after use for several consecutive reproductive cycles [39,40,41]. In addition, even with a more efficient microbiota, the feed restriction during gestation, and consequently of the substrate for intestinal bacteria, may have limited the piglets’ birth weight from sows that consumed fiber in the final third of gestation [42].

Piglets whose mothers received fiber supplementation in both phases were weaned heavier than piglets in the other groups. This may be related to the increased milk production promoted by the higher feed intake. In addition, the modulation of the intestinal microbiota resulted in high concentrations of SCFAs from fiber fermentation in the gut, which are absorbed and used as an energy source [43]. When passing from a restricted feeding system (gestation) to a high food supply (lactation), these sows produce a great amount of SCFAs due to the high availability of substrate for bacteria. The high absorption of short-chain fatty acids and triglycerides contributes to fat retention in the mammary glands [16]. The estimated milk dry matter production tended to be higher after 21 days of lactation in sows with high-fiber supplementation during the last third of gestation compared to sows without high-fiber supplementation, which could also favor better piglet development [16].

In addition, newborn piglets from sows supplemented with high-fiber during gestation present intestinal maturation and crypt depth [44,45]. This contributes to establishing their microbiota at birth, increasing their development potential. A smaller amount of undesirable bacteria in the environment where the piglets are inserted contributes to reducing the occurrence of piglets with diarrhea in maternity [46]. Fiber intake by sows during gestation affects the intestinal microbial modulation of piglets at 14 days of age, playing an important role in their immune system and body metabolism [47,48]. The gastrointestinal bacterial colonization of newborns is originated from the maternal intestine, vagina, amniotic fluid, and placenta and may occur in the prenatal period [49]. This bacterial profile is influenced by diet, antibiotic exposure, stress, and sow health [49]. The dietary fiber supplementation during gestation increases the acetate level in the digesta, which is crucial for developing T cells and improving the piglet’s intestinal function [50].

Comparing the results of piglets’ weaning weight from sows that did not receive fiber in any of the phases (CC) with those whose mothers were supplemented with fiber during late pregnancy and lactation (FF), the difference in average weight at weaning was 1.255 kg. The commercial unit where this current study was carried out has 2869 sows with an average of 2.5 farrowing episodes per year and 12 weaned piglets per farrowing. The difference in weight gain during lactation results in more than 108,000 kg of piglets weaned yearly due to the supplementation of eubiotic dietary fiber for sows.

When evaluating the cost–benefit of using dietary fiber during gestation and/or lactation, a more advantageous scenario occurred when sows received supplementation in both phases. For example, for a piglet production unit with 2000 sows, 2.5 farrowing/sow/year, and an average of 12 weaned piglets/farrowing with an average weight of 6.0 kg, there are 360,000 kg of piglets weaned per year. Considering the difference of USD 0.15 in the cost per kg of piglet produced between the CC and FF treatments, the savings in favor of groups that received fiber in both phases was around USD 54,000 per year.

Calculating the revenue obtained from sales, there is a financial advantage of USD 26.68 per litter of sows that received fiber in both phases (FF) compared to those that did not receive fiber in either treatment (CC). For a UPL with approximately 5000 births per year, it would represent an additional income of USD 133,400 per year.

### 4.2. Surface Temperature of Sows

Sows supplemented with eubiotic fiber during gestation had a higher body temperature during lactation compared to sows that did not receive fiber. This could be associated with an increased feed intake and, consequently, a high caloric increase [51]. Sows that received eubiotic dietary fiber in both phases had a higher temperature of mammary glands than those that received fiber only during gestation. The great supply of energy and nutrients to mammary glands resulting from the high consumption of feed during lactation and the possible modulation of the microbiota during the final third of gestation (promoting great production of short-chain fatty acids and better digestibility and use of the diet) [52] may result in a great milk production, which could partially justify the better performance of litters from the sows of this treatment. In turn, the great supply of nutrients to the mammary glands, with a consequent increase in metabolism and milk production, leads to an increase in tissue temperature.

### 4.3. Behavioral Assessments

During gestation, sows that received eubiotic dietary fiber spent more time lying ventrally. Additionally, they spent less time sprouting around the pen components and, in addition, reduced the frequency of stereotyped behaviors such as floor licking, false chewing, and agonistic behaviors. Feed restriction for gestating sows aims to avoid excessive weight gain at this stage [53]. However, excessive feeding restrictions make sows unsatisfied, leading to a frequent occurrence of stereotyped behaviors and fighting when housed in collective pens [54]. Thus, considering the capacity of modulation of glycemic peaks, including insoluble fibers in the diet during gestation can be considered a good nutritional strategy to promote satiety and reduce the apparent motivation in feeding sows without providing excess energy [19,40,55]. Cassar et al. [56] observed that sows in gestation, under the effects of feed restriction, increased the time they spent lying down by being supplemented with fibrous feed.

The reduction of standing activity has been observed with the incorporation of fibrous components in the diet of these animals. The number of postural changes also decreased with the supplementation of fibrous diets. There was a strong effect when sows fed on a diet based on oat hulls compared to wheat bran and corn cob [57]. Che et al. [41] observed a beneficial effect of the incorporation of a high content of plant cell wall in the gestation diet of multiparous sows expressed by a reduction in the time spent in the standing position and an increase in the lying position.

Sows that received fiber during lactation, whether or not they received it during gestation, spent less time sitting, in relation to the sows that were not supplemented during this period. Sitting or standing inactive for long periods may indicate poor welfare. On the other hand, the lying position may reflect a welfare situation in the case of sows housed in collective pens [58].

Animals that received fiber during the last third of gestation and lactation had a higher frequency of visits to the drinking fountain than those that received fiber only during lactation. The animal’s daily water requirement determined its access to the drinker. The high frequency of access to the drinker may be related to the increase in feed consumption by sows supplemented with fiber in both phases; for example, solid feed intake must be accompanied by water intake [59], especially in diets with higher fiber content due to its solubility and water holding capacity [60].

Stereotyped behaviors also decreased during lactation in sows that received fiber during this period. However, this reduction was more significant in sows that had already been supplemented since the final third of gestation. At this stage, stereotypes are not related to food restriction but to the severe restriction of movement imposed by farrowing units.

The gut microbiota affects the functioning of the gut–brain axis and the passage of metabolites and neurotransmitters produced by the gut [61], which affects neural circuits and behaviors associated with a stressful response [62]. Studies by [63,64] demonstrated that humans with depression have lower diversity in the gut microbiota and high levels of inflammatory markers. According to [65], patients with inflammatory diseases of the gastrointestinal tract often have anxiety and depression, possibly due to dysregulations in tryptophan metabolism and the consequent production of serotonin.

However, sows that received fiber during gestation but did not receive fiber during lactation showed high levels of stereotypies, demonstrating the momentary effect of fiber. According to [66], stereotyped behaviors may be more apparent when the environment is restricted and mitigated with increased diet fiber and more frequent feeding. Bergeron et al. [67] observed that a high-fiber diet (29% ADF, 50% NDF) promoted a shortening in the time spent stereotyping by sows within two hours after a meal.

## 5. Conclusions

The use of eubiotic fiber, insoluble and partially fermentable, in diets during the final third of pregnancy and lactation promotes improvements in the well-being of sows, contributing to a shorter duration of farrowing, greater feed consumption during lactation, greater weight gain of their piglets, and reduction in stereotyped behaviors of females in both phases.

## Figures and Tables

**Figure 1 animals-13-00695-f001:**
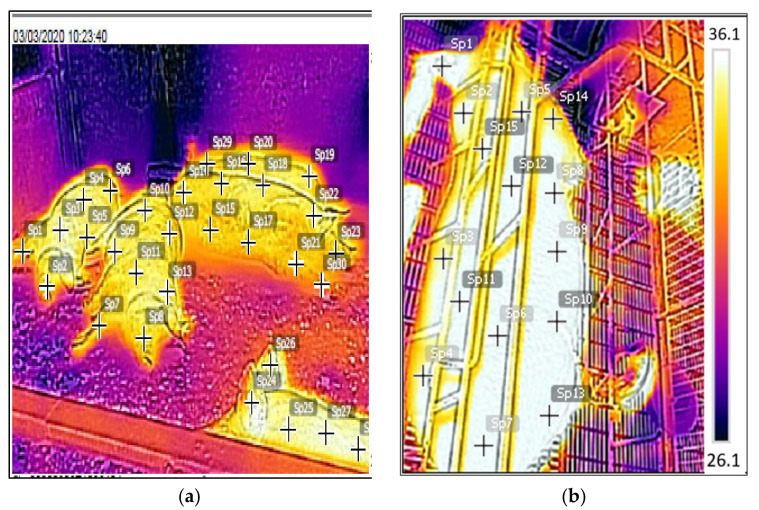
(**a**) Thermographic image of sows in collective gestation, (**b**) thermographic image of a lactating sow.

**Table 1 animals-13-00695-t001:** Percentage and calculated nutritional composition of experimental diets during the phases of gestation and lactation.

Percentage Composition		
Ingredient	Gestation	Lactation
Corn	80.33	62.07
Soybean meal 46%	15.70	28.85
Soy oil	—	5.20
Limestone	1.45	1.29
Monodicalcium Phosphate	1.41	0.98
Salt	0.60	0.55
Mineral premix ^1^	0.15	0.15
Vitamin premix ^2^	0.05	0.05
Liquid Choline Chloride 75%	0.14	0.11
Liquid lysine	0.12	0.47
Methionine MHA 88%	—	0.15
Threonine	0.04	0.12
Phytase	0.005	0.005
Carbohydrase	—	0.005
Calculated Nutritional Composition		
Metabolizable energy (kcal/kg)	3240.4	3510.2
Crude protein (%)	14.95	19.92
Digestible Lysine (%)	0.72	1.25
Digestible Methionine (%)	0.23	0.30
Digestible Threonine (%)	0.45	0.71
Digestible Tryptophan (%)	0.49	0.24
Crude fiber (%)	1.86	2.16
Calcium (%)	0.75	0.80
Available phosphorus (%)	0.38	0.38
Sodium (%)	0.20	0.23

^1^ Composition per kg of product: Fe, 180 g; Cu, 20 g; Co, 4 g; Mn, 80 g; Zn, 140 g; 1.4 g and excipient qsp, 1000 g. ^2^ Composition per kg of product: vit. A, 12,000,000 IU; vit. D3, 12,000,000 IU; vit. E, 8000 IU; vit. K3, 4 g; vit. B2, 4 g; vit. B6, 5 g; vit. B12, 30,000 mg; nicotinic acid, 40 g; pantothenic acid, 20 g; zinc bacitracin, 10 g; antioxidant, 30 g; selenium, 23 mg; excipient qsp, 1000 g.

**Table 2 animals-13-00695-t002:** Ethogram used for behavioral evaluation of sows in the final third of gestation.

Category	Behavior	Description
Position	Lying sideways	Lying with one shoulder in contact with the floor
	Lying ventrally	Lying with the chest and abdomen in contact with the floor
	Standing	Standing on four limbs
	Kneeling	Supported on the carpi with stretched hind limbs
	Seated	In sitting position
Activity	Walking	Moving around the pen
	Eating	Eating the offered feed
	Drinking	Drinking water
	Scouring around	Scouring the floor of pens
Behavior agonistic	Interacting negatively	Interacting negatively with other females in the pen (pushing, chasing, threatening, and biting)
Behavior social	Interacting positively	Positively interacting with other females (including nibbling, sniffing, and licking body parts)
Behavior stereotyped	Licking the floor/rails	Licking the floor or repeatedly biting the rails of pens
	False chewing	Repetitive movement of the tongue, simulating chewing, but with no food

**Table 3 animals-13-00695-t003:** Ethogram used for behavioral evaluation of sows in the lactation phase.

Category	Behavior	Description
Position	Lying sideways	Lying with one shoulder in contact with the floor
	Lying ventrally	Lying with the chest and abdomen in contact with the floor
	Standing	Standing on four limbs
	Kneeling	Supported on the carpi with stretched hind limbs
	Seated	In sitting position
Activity	Breastfeeding	Lying on the side suckling the piglets
	Eating	Eating the offered feed
	Drinking	Drinking water
Behavior stereotyped	Biting cage or equipment	Biting or sniffing the iron bars, floor, feeder, and drinker without eating or drinking water
	False chewing	Repetitive movement of the tongue, simulating chewing, but with no food

**Table 4 animals-13-00695-t004:** Weight and condition score at discharge (Initial Weight and BCS) from farrowing, average lactation feed intake (ADFI), and weaning-to-estrus interval (WEI) of sows supplemented or not with dietary fiber during gestation and/or lactation.

Variable	Gestation	Lactation		Mean	SEM	*p*-Value		
		Control	Fiber			Gestation	Lactation	G × L
Initial Weight (kg)	Control	238.91	244.70	241.81	0.14	0.233	0.147	0.536
	Fiber	243.98	246.31	245.14				
	Mean	241.45	245.50	243.48				
ADFI	Control	172.01 Ba	169.10 Ba	170.55	0.77	<0.001	0.836	0.003
	Fiber	195.06 Aa	197.64 Aa	196.35				
	Mean	183.53	183.37	183.45				
INITIAL BCS	Control	2.68	2.69	2.68	0.02	0.467	0.046	0.059
	Fiber	2.57	2.71	2.66				
	Mean	2.63 b	2.72 a	2.67				
WEI (days)	Control	4.63	5.19	4.91	0.21	0.259	0.636	0.725
	Fiber	4.55	4.31	4.43				
	Mean	4.59	4.75	4.67				

Means without a common uppercase letter (A–B) in the same column and a common lowercase letter (a–b) in the same row differed at *p* < 0.05.

**Table 5 animals-13-00695-t005:** Number of piglets per sow after litter standardization (LN), weight of piglets 48 h after birth (W 48 h), number of weaned piglets per sow, weight of piglets at weaning, coefficient of variation of weight at weaning, and mortality rate of piglets from sows supplemented with dietary fiber during gestation and/or lactation.

Variable	Gestation	Lactation		Mean	SEM	*p*-Value		
		Control	Fiber			Gestation	Lactation	G × L
LN	Control	13.90	13.91	13.91	0.06	0.509	0.805	0.760
	Fiber	14.01	13.95	13.98				
	Mean	13.96	13.93					
W 48 h (kg)	Control	1.65	1.66	1.66	0.02	0.611	0.841	0.947
	Fiber	1.63	1.64	1.64				
	Mean	1.64	1.65					
Weaned piglets (n)	Control	12.50	12.51	12.50	0.07	0.489	0.877	0.902
	Fiber	12.58	12.62	12.60				
	Mean	12.54	12.56					
Weight at weaning (kg)	Control	4.92 Ab	5.26 Ba	5.09	0.04	< 0.001	<0.001	<0.001
	Fiber	5.0 2 Ab	6.17 Aa	5.60				
	Mean	4.97	5.715					
CV weight at weaning (%)	Control	17.49	17.73	17.61	0.02	0.187	0.298	0.119
	Fiber	17.60	16.38	16.99				
	Mean	17.55	17.06					
Mortality (%)	Meal	0.73	0.69	0.71	0.06	0.348	0.399	0.512
	Fiber	0.97	0.77	0.87				
	Mean	0.85	0.73					

Means without a common uppercase letter (A–B) in the same column and a common lowercase letter (a–b) in the same row differed at *p* < 0.05.

**Table 6 animals-13-00695-t006:** Number of piglets born alive, stillbirths, average piglet weight at birth, litter uniformity (coefficient of variation of birth weight and piglets weighing less than 1.0 kg at birth), farrowing duration, weight, and BCS of sows entering the farrowing unit supplemented or not with dietary fiber during gestation.

Variable (%)	Gestation		SEM	*p*-Value
	Control	Fiber		
Live births (n)	15.73	15.36	0.16	0.245
Stillbirths (n)	0.98	1.06	0.06	0.915
Birth weight (kg)	1.33	1.34	0.01	0.505
CV birth weight (%)	21.43	21.58	0.01	0.826
Piglets < 1.0 kg at birth (n)	2.93	2.87	0.14	0.831
Duration of farrowing (min)	300.47	260.55	0.04	0.006
Initial weight of sows (kg)	268.77	272.13	2.14	0.257
Initial BCS of sows	2.79	2.79	0.02	0.908

**Table 7 animals-13-00695-t007:** Body surface temperature (MST) and mammary system surface temperature (MSST); temperature means (°C) of lactating sows with or without supplementation of dietary fiber in the diet.

Variable	Gestation	Lactation		Mean	SEM	*p*-Value		
		Control	Fiber			G	L	G × L
MST	Control	34.89	34.65	34.77 B	0.04	0.021	0.053	0.160
	Fiber	34.96	34.92	34.94 A				
	Mean	34.93	34.78	34.85				
MSST	Control	34.62 Aa	34.78 Aa	34.70	0.04	0.273	<0.001	0.017
	Fiber	34.35 Ab	34.88 Aa	34.61				
	Mean	34.48	34.83	34.66				

Means without a common uppercase letter (A–B) in the same column and a common lowercase letter (a–b) in the same row differed at *p* < 0.05.

**Table 8 animals-13-00695-t008:** Mean body surface temperature (°C) in the morning (MST-M) and the afternoon (MST-A); mean temperature (°C) of pregnant sows with or without supplementation of dietary fiber in the diet.

Variable (%)	Gestation		SEM	*p*-Value
	Control	Fiber		
MST-M	32.35	32.15	0.19	0.566
MST-A	33.07	33.28	0.20	0.610

**Table 9 animals-13-00695-t009:** Behavioral frequency (%) of sows in collective gestation (85th to 107th days of gestation) supplemented or not with dietary fiber in the diet.

Behavior (%)	Gestation		SEM	*p*-Value
	Control	Fiber		
Lying sideways	16.72	16.68	0.05	0.579
Lying ventrally	10.75 b	13.18 a	0.06	0.020
Seated	5.75	7.53	0.09	0.124
Kneeling	12.78	12.74	0.09	0.602
Standing	4.15	4.70	0.07	0.769
Walking	14.62	16.63	0.08	0.660
Eating	3.87	4.13	0.08	0.678
Drinking	7.78	7.00	0.09	0.203
Scouring	6.57	6.07	0.08	0.051
Interacting negatively	4.66	3.46	0.13	0.064
Interacting positively	5.24 a	3.73 b	0.13	0.011
Licking the floor	4.75 a	2.93 b	0.12	0.001
False chewing	2.36 a	1.19 b	0.11	<0.001

Means without a common lowercase letter (a–b) in the same row differed at *p* < 0.05.

**Table 10 animals-13-00695-t010:** Behavioral frequency (%) of sows during the stay in farrowing units (seven days before farrowing to weaning) supplemented or not with dietary fiber during gestation and lactation.

Variable	Gestation (G)	Lactation (L)		Mean	SEM	*p*-Value		
		Control	Fiber			Gestation	Lactation	G × L
Lying sideways	Control	25.47 Aa	27.47 Ba	26.47	0.02	0.779	<0.001	<0.001
	Fiber	24.19 Ab	29.41 Ba	26.80				
	Mean	24.83	28.44					
Lying ventrally	Control	18.86	21.30	20.08 B	0.02	0.002	<0.001	0.057
	Fiber	20.77	22.19	21.48 A				
	Mean	19.81 b	21.75 a					
Seated	Control	4.49	1.80	3.15	0.07	0.739	<0.001	0.462
	Fiber	4.04	1.15	2.60				
	Mean	4.27 a	1.48 b					
Kneeling	Control	0.06	0.00	0.03	0.00	1.000	1.000	1.000
	Fiber	0.00	0.00	0.00				
	Mean	0.03	0.00					
Standing	Control	20.79	19.81	20.30	0.03	0.413	0.243	0.058
	Fiber	19.93	21.70	20.82				
	Mean	20.36	20.75					
Breastfeeding	Control	6.47	7.80	7.14	0.02	0.406	<0.001	0.397
	Fiber	6.44	8.21	7.32				
	Mean	6.45 b	8.01 a					
Eating	Control	5.25	5.92	5.59 b	0.01	0.002	<0.001	0.099
	Fiber	5.38	6.59	5.98 a				
	Mean	5.32 b	6.26 a					
Drinking	Control	5.07 Aa	4.69 Ba	4.88	0.02	0.045	0.516	0.023
	Fiber	4.97 Aa	5.32 Aa	5.15				
	Mean	5.02	5.01					
Biting rails	Control	6.48 Aa	5.50 Aa	5.99	0.04	0.093	<0.001	0.003
	Fiber	7.28 Aa	3.23 Bb	5.25				
	Mean	6.88	4.37					
Interacting positively	Control	0.00	0.00	0.00	0.00	n/a	n/a	n/a
	Fiber	0.00	0.00	0.00				
	Mean	0.00	0.00					
False chewing	Control	7.06 Aa	5.69 Aa	6.37	0.04	0.005	< 0.001	<0.001
	Fiber	6.70 Aa	2.24 Bb	4.62				
	Mean	7.03	3.96					

Means without a common uppercase letter (A–B) in the same column and a common lowercase letter (a–b) in the same row differed at *p* < 0.05.

**Table 11 animals-13-00695-t011:** Cost–benefit analysis of using dietary fiber (Eubiotic fiber) with sows in gestation and/or lactation.

	CC	CF	FC	FF
Consumption of gestation food (kg) ^1^	48.40	48.40	48.40	48.40
Consumption of lactation feed (kg) ^2^	172.01	169.10	195.06	197.64
Eubiotic fiber consumption (kg) ^3^	0.00	1.48	1.21	2.69
Eubiotic fiber cost/female (USD)	0.00	1.76	1.44	3.20
Cost with food/female (USD) ^4^	115.34	113.75	127.80	129.39
Total cost/female (food + Eubiotic fiber)	115.34	115.51	129.24	132.59
No. of weaned piglets	12.58	12.62	12.54	12.56
Weight at weaning (kg)	4.92	5.26	5.02	6.17
Cost/kg of weaned piglet (USD) ^5^	1.86	1.74	2.05	1.71
Revenue per litter (USD) ^6^	175.42	188.21	178.69	219.86
Profit per litter (USD) ^7^	60.08	72.70	49.45	87.27

^1^ Consumption at 22 days (85th to 107th day of gestation) (2.2 kg/day). ^2^ Consumption in 27 days (20 days of lactation + 7 days before farrowing). ^3^ Gestation: 55 g/day × 22 days = 1210 g; Lactation: 55 g/day × 27 days = 1485 g. ^4^ Total cost of gestation and lactation feed (consumption × cost/kg). ^5^ Total cost/sow ÷ Kg of weaned piglets/litter (weaned piglet’s × weight at weaning). ^6^ Kg of weaned piglets per litter × price per kg of weaned piglets (USD 2.83). ^7^ Revenue per litter—Total cost/female.

## Data Availability

Not applicable.

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
