# Peer review of "Dietary Supplementation of Eubiotic Fiber Based on Lignocellulose on Performance and Welfare of Gestating and Lactating Sows"

_animals, 2023, doi:10.3390/ani13040695_

Round 1

Reviewer 1 Report

It is an interesting and complete research work. Some concepts and paragraphs must be clarified as well as in material and methods, to specify the statistical tests done for the behavioral observations to improve the article.

Reviewer 2 Report

Dear Authors,

Thank you for the interesting paper. However, some areas need improvements before this can be considered for publication. For instance, you have ignored basic author guidelines when citing the references and numbering the headings and subheadings. In addition, some basic formatting errors can be seen throughout the manuscript. For instance, improper spaces between words, unnecessary spaces between words, sentence fragments, not following the correct order for tables within the text, the incorrect numbering of tables within the text...etc., can not be accepted at this level. One of the major drawbacks is the hypothesis that you mentioned. I can not see any match between the hypothesis and the objective of your study. So, please address those carefully. Other comments are also attached. Good luck.  

Author Response

Por favor, verifique o anexo.

Reviewer 3 Report

Authors Agnês Markiy Odakura et al. reported the effects of partially fermentable insoluble dietary 20 fiber supplementation on the behavior, surface temperature, and reproductive parameters of ges- 21 tating and lactating sows and found that dietary supplementation of eubiotic fiber for sows in end period of gestation and lactation im- 32 proved performance and welfare, with positive consequences for developing their litters. It was a very sample study maybe better submit to a local journal. Also the language need carefully revised.

1. There are many unnecessary spaces.

2. P value shou be p.

3. The introduction and discussion parts were both insufficent.

Round 2

Reviewer 2 Report

Thank you for the revised version of the manuscript; it has improved. However, I still see some spacing issues throughout the text (e.g., Lines 26-30). So, please pay attention to those and correct them. In addition, please check for the "p" letter, including the Tables. 

Reviewer 3 Report

It could be accepted in the current version.